# Effects of One Night of Forced Wakefulness on Morning Resting Blood Pressure in Humans: The Role of Biological Sex and Weight Status

Lieve T. van Egmond [1,2,*] , Pei Xue [2], Elisa M. S. Meth [2], Maria Ilemosoglou [2] , Joachim Engström [2] and Christian Benedict [2]

1   Department of Surgical Sciences, Uppsala University, Box 593, 751 24 Uppsala, Sweden
2   Department of Pharmaceutical Biosciences, Uppsala University, Box 593, 751 24 Uppsala, Sweden
*   Correspondence: lieve.van.egmond@neuro.uu.se

**Abstract:** Permanent night shift work is associated with adverse health effects, including elevated blood pressure (BP) and hypertension. Here, we examined the BP response to one night of forced wakefulness in a sitting position in a cohort without night shift work experience. According to a counterbalanced crossover design, 47 young adults with either obesity (N = 22; 10 women) or normal weight (N = 25; 11 women) participated in one night of sleep and one night of forced wakefulness under in-laboratory conditions. Resting ankle and brachial arterial BP were assessed in the morning, i.e., the time of the day when adverse cardiovascular events peak. After forced wakefulness, diastolic and mean arterial BP were ~4 mmHg higher at the ankle site and ~3 mmHg higher at the brachial site than after regular sleep ($p < 0.05$). The increase in BP following overnight forced wakefulness was more pronounced among men vs. women and more significant for diastolic BP at both sites among participants with normal weight vs. those with obesity. If confirmed in larger cohorts, including 24 h BP monitoring, people with occupations involving night shifts might benefit from regular BP monitoring. Particular attention should be paid to possible sex- and weight-specific effects of night shift work on BP.

**Keywords:** sleep deprivation; blood pressure; sex differences; obesity

## 1. Introduction

The COVID-19 pandemic has clearly emphasized the importance of around-the-clock access to essential services, such as medical care. One of the main pillars of running a society 24/7 is night shift work, carried out occasionally or permanently by about 20% of the workforce. However, long-term night shift work is associated with adverse health effects, including elevated blood pressure (BP) and hypertension. For example, a recent meta-analysis based on 45 studies involving 117,252 workers found that permanent night workers had a ~2.5 mmHg higher systolic BP and ~2 mmHg higher diastolic BP compared to daytime workers [1]. Furthermore, in a cross-sectional study from China involving over 80,000 female nurses, working at least five nights per month increased hypertension risk by 19% to 32% [2]. Finally, a recent study based on data from the UK Biobank demonstrated that night shift workers had a higher hypertension risk than day shift workers, which increased with the increasing frequency of night shift work [3]. Blood pressure follows a 24 h rhythm, with a marked drop during nighttime sleep and a steep increase in the morning around awakening [4]. Not coincidentally, most major adverse cardiovascular events occur during morning hours [5]. Hence, it is of particular interest to investigate whether night shift work, even in the short term, is associated with higher BP values in the morning.

Apart from night shift work, the male sex has been associated with higher BP [6]. For example, studies using 24 h ambulatory BP monitoring have shown that BP is higher in men

than in women at similar ages [7]. Another risk factor for elevated BP is excess fat stored around the abdomen. In the Olivetti Heart Study, waist circumference, an index of central adiposity, was a strong predictor of higher diastolic and systolic BP values. This association was also seen when controlling for other metabolic perturbations, e.g., insulin resistance [8]. Noteworthy, whether the BP response to one night of forced wakefulness differs between men and women and varies by a person's weight status is largely unknown, despite certain occupations involving night shifts being predominated by women (e.g., nurses) and the fact that the worldwide prevalence of obesity nearly tripled between 1975 and 2016 [9].

In the present crossover study, including men and women with normal weight and obesity, we measured supine BP after a 15 min rest, in the morning after one night of sleep and one night of forced wakefulness, mainly spent in a sedentary position. The latter condition primarily represents sitting occupations involving night shifts (e.g., hotel receptionists).

Given that both the male sex and obesity have been associated with higher BP [6–8], we hypothesized that the BP response to one night of forced wakefulness would be more pronounced among men and participants with obesity. Since high ankle BP has been associated with mortality [10], and higher brachial BP increases the risk of fatal and non-fatal cardiovascular events [11], we assessed resting BP at both anatomical sites.

## 2. Results

### 2.1. Cohort Characteristics

A total of 508 potential participants were screened for possible study inclusion. However, of the 508 people, 461 were not considered eligible, e.g., they had a chronic disease, did not use hormonal contraceptives (for women), smoked, had an extreme chronotype, or did not respond. In addition, we included only participants without past employment involving night or rotating shifts to reduce possible confounding due to differences in the ability to cope with one night of forced wakefulness.

Following these exclusions, 47 participants remained for the present study. The included men (N = 26) and women (N = 21) did not significantly differ in age (mean $\pm$ SEM, men vs. women, 25 $\pm$ 3 vs. 25 $\pm$ 3 years, $p = 0.631$), BMI (28.0 $\pm$ 6.7 vs. 27.2 $\pm$ 5.9 kg/m$^2$, $p = 0.674$), waist circumference (95 $\pm$ 17 vs. 88 $\pm$ 17 cm, $p = 0.187$), and chronotype (assessed by the Morningness Eveningness Questionnaire (MEQ; [12]); 53 $\pm$ 8 vs. 51 $\pm$ 9 points, $p = 0.552$). The weight groups (i.e., those with normal weight, N = 25, vs. those with obesity, N = 22) were comparable for age (25 $\pm$ 3 vs. 25 $\pm$ 3 years, $p = 0.731$), chronotype (53 $\pm$ 10 vs. 51 $\pm$ 6 points, $p = 0.295$) but not BMI (22.6 $\pm$ 2.0 vs. 33.9 $\pm$ 3.3 kg/m$^2$, $p < 0.001$) and waist circumference (78.3 $\pm$ 8.2 vs. 108.2 $\pm$ 9.1 cm, $p < 0.001$).

### 2.2. Main Effects of Experimental Condition, Biological Sex, and Weight Group Status on Morning Resting BP

Following the night of forced wakefulness, participants' morning diastolic and mean arterial BP was elevated by 4.2 mmHg and 3.6 mmHg at the ankles and by 2.7 mmHg and 2.8 mmHg at the arms, respectively ($p \leq 0.05$; Table 1 and Figure 1). In contrast, the rise in systolic BP following forced overnight wakefulness was only descriptively different compared to values measured after sleep (Table 1 and Figure 1). Finally, all studied BP outcomes, except for the ankle systolic and mean arterial BP, were significantly higher among men than women (Table 1). In addition, the majority of BP metrics among participants with obesity significantly exceeded those of participants with normal weight (Table 1).

**Table 1.** Association of participants' biological sex, weight group status, and experimental condition with morning resting blood pressure.

| Outcome | Experimental Condition (Co) | | Biological Sex (Sex) | | Weight Group (W) | | p for Interaction | |
|---|---|---|---|---|---|---|---|---|
| | Wakefulness | Sleep | Men | Women | Obesity | Normal Weight | Co*Sex | Co*W |
| No. of participants | 47 | 45 | 26 | 21 | 22 | 25 | – | – |
| No. of women | 21 | 21 | 0 | 21 | 10 | 11 | – | – |
| No. of participants with obesity | 22 | 20 | 12 | 10 | 22 | 0 | – | – |
| **Ankle site** | | | | | | | | |
| Systolic BP, mmHg | 132.9 ± 9.9 | 130.8 ± 9.8 | 133.5 ± 9.9 | 130.2 ± 9.9 | 134.9 ± 9.9 * | 128.8 ± 9.9 | 0.019 | 0.470 |
| Diastolic BP, mmHg | 69.4 ± 5.9 * | 65.2 ± 5.8 | 69.4 ± 5.9 ** | 65.3 ± 5.9 | 68.5 ± 5.9 | 66.1 ± 5.9 | 0.076 | 0.567 |
| Mean arterial BP, mmHg | 90.7 ± 6.2 * | 87.1 ± 6.1 | 90.5 ± 6.1 | 87.3 ± 6.2 | 90.8 ± 6.2 * | 86.9 ± 6.1 | 0.018 | 0.476 |
| **Brachial site** | | | | | | | | |
| Systolic BP, mmHg | 115.9 ± 6.9 | 112.9 ± 6.9 | 117.1 ± 6.9 * | 111.7 ± 7.0 | 117.3 ± 7.0 ** | 111.5 ± 6.9 | 0.245 | 0.299 |
| Diastolic BP, mmHg | 68.2 ± 4.5 * | 65.5 ± 4.4 | 68.8 ± 4.5 ** | 65.0 ± 4.5 | 68.9 ± 4.5 *** | 64.9 ± 4.4 | 0.453 | 0.564 |
| Mean arterial BP, mmHg | 84.1 ± 4.6 * | 81.3 ± 4.6 | 84.9 ± 4.6 *** | 80.6 ± 4.6 | 85.0 ± 4.6 *** | 80.5 ± 4.6 | 0.298 | 0.927 |

Estimated marginal means ± standard error and *p*-values derived from generalized linear mixed models, assuming an unstructured covariance matrix and using a normal distribution with an identity link function. * $p < 0.05$, ** $p < 0.001$, and *** $p \leq 0.001$ for pairwise contrasts. Abbreviations: BP, blood pressure; Co, experimental condition; mmHg, millimeter mercury; No., number; W, weight group.

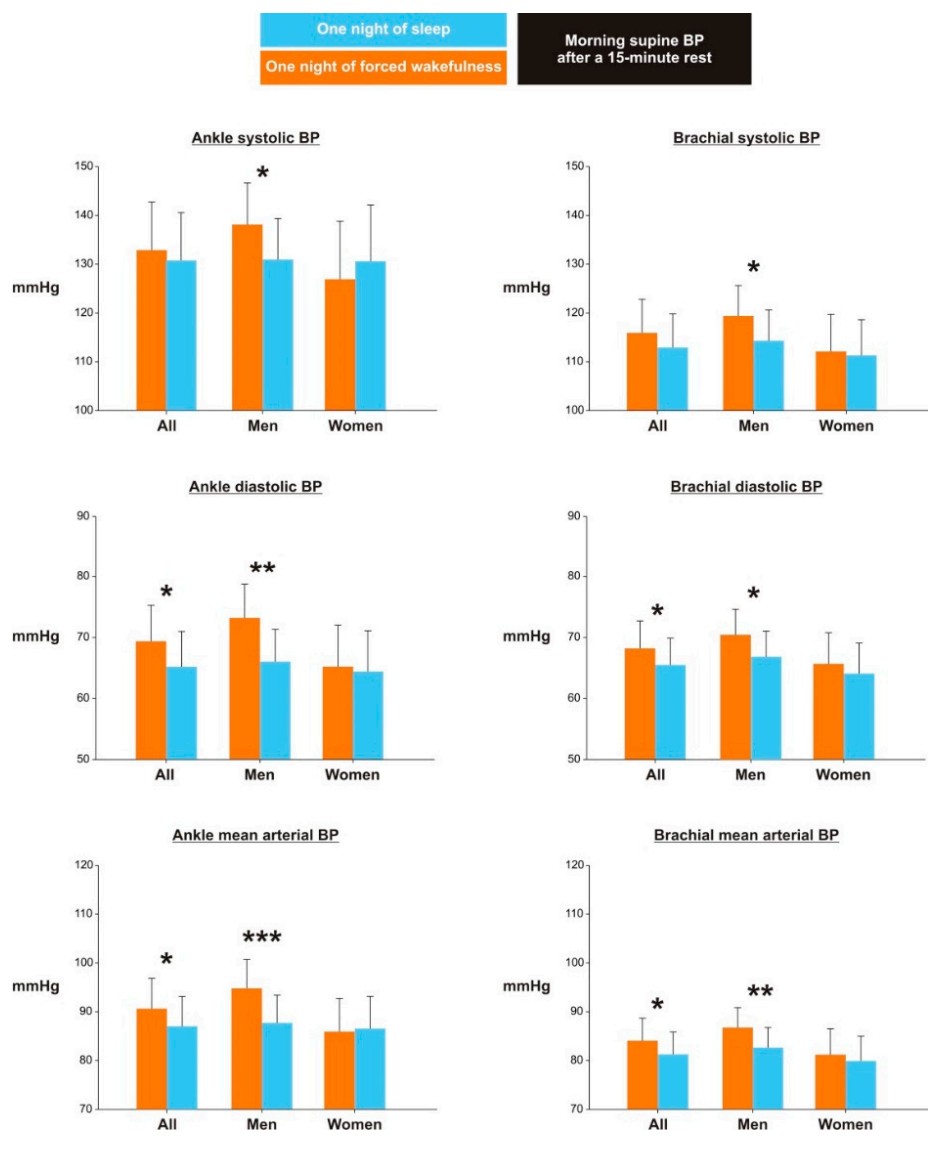

**Figure 1.** Effects of overnight wakefulness on arterial blood pressure. Estimated marginal means ± standard error and *p*-values derived from generalized linear mixed models, assuming an unstructured covariance matrix and using a normal distribution with an identity link function. Orange: overnight wakefulness condition, blue: sleep condition. * $p < 0.05$, ** $p < 0.001$, and *** $p \leq 0.001$ for pairwise contrasts. Abbreviations: BP, blood pressure; mmHg, millimeter mercury.

## 2.3. Sex-Specific Effects of Forced Overnight Wakefulness on BP

The effects of forced overnight wakefulness on ankle BP differed between men and women (*p*-values for the interaction between the experimental condition and participants' sex are shown in Table 1). Although ankle systolic, diastolic, and mean arterial BP rose in men following forced wakefulness (+7.2 mmHg for systolic BP, $p < 0.05$; +7.2 mmHg for diastolic BP, $p \leq 0.001$; and +7.1 mmHg for mean arterial BP, $p < 0.01$; Table 2 and Figure 1), no significant differences in ankle BP outcomes were found between the experimental conditions in women (Table 2 and Figure 1). Although similar sex-specific effects of forced overnight wakefulness on morning brachial BP values emerged (i.e., men had overall higher branchial BP values than women after the night of forced wakefulness; Table 2), no significant interactions between participants' biological sex and experimental condition were found (Table 1).

**Table 2.** Effects of one night of forced wakefulness on morning blood pressure, split by participants' sex and weight group status.

| Outcome | Men (§) | | Women (§) | | Normal Weight (†) | | Obesity (†) | |
|---|---|---|---|---|---|---|---|---|
| | Wakefulness | Sleep | Wakefulness | Sleep | Wakefulness | Sleep | Wakefulness | Sleep |
| **Ankle site** | | | | | | | | |
| Systolic BP, mmHg | 138.1 ± 8.5 * | 130.9 ± 8.4 | 126.9 ± 11.9 | 130.6 ± 11.5 | 130.8 ± 7.1 | 127.0 ± 7.0 | 134.7 ± 13.6 | 134.4 ± 13.1 |
| Diastolic BP, mmHg | 73.2 ± 5.5 *** | 66.0 ± 5.3 | 65.2 ± 6.8 | 64.4 ± 6.7 | 68.8 ± 6.4 * | 63.6 ± 6.1 | 69.7 ± 5.8 | 67.2 ± 5.9 |
| Mean arterial BP, mmHg | 94.8 ± 5.9 *** | 87.7 ± 5.7 | 85.9 ± 6.8 | 86.5 ± 6.6 | 89.5 ± 6.0 * | 84.8 ± 5.7 | 91.6 ± 7.1 | 89.3 ± 6.9 |
| **Brachial site** | | | | | | | | |
| Systolic BP, mmHg | 119.4 ± 6.2 * | 114.3 ± 6.3 | 112.1 ± 7.6 | 111.3 ± 7.3 | 111.8 ± 6.8 | 110.4 ± 6.8 | 120.2 ± 6.9 | 115.0 ± 6.9 |
| Diastolic BP, mmHg | 70.5 ± 4.2 * | 66.9 ± 4.2 | 65.7 ± 5.1 | 64.1 ± 5.0 | 66.6 ± 4.0 * | 63.2 ± 3.9 | 69.8 ± 5.3 | 67.8 ± 5.3 |
| Mean arterial BP, mmHg | 86.8 ± 4.1 ** | 82.7 ± 4.1 | 81.2 ± 5.3 | 79.9 ± 5.1 | 81.6 ± 4.3 | 78.9 ± 4.2 | 86.5 ± 5.2 | 83.7 ± 5.1 |

Data are reported as estimated marginal means ± standard error. *p*-values derived from generalized linear mixed models (GLMMs) assuming an unstructured covariance matrix and using normal distribution with an identity link function. § = GLMMs included participants' weight group status and experimental condition as fixed factors and subject-ID as a random factor; and † = GLMMs included participants' biological sex and experimental condition as fixed factors and subject-ID as a random factor. * $p < 0.05$, ** $p < 0.01$, and *** $p \leq 0.001$ for pairwise contrasts. Abbreviations: BP, blood pressure; mmHg, millimeter mercury.

## 2.4. Weight Group-Specific Effects of Forced Overnight Wakefulness on BP

At first glance, the effects of forced wakefulness on morning resting diastolic BP appeared to be more pronounced at both the brachial and ankle sites among those with normal weight (Table 2). However, none of the interactions between the experimental condition and participants' weight group status reached significance (Table 1).

As sleep-disordered breathing is common in people with obesity and could severely impact sleep and BP [13,14], we applied linear regression adjusting for sex and weight status to examine whether the number of oxygen desaturation events per hour (measured by the oxygen desaturation index (ODI); [15]) is associated with morning BP outcomes in the sleep condition. However, ODI did not significantly predict morning BP in the sleep condition (ankle site: $p = 0.877$ for systolic BP; $p = 0.580$ for diastolic BP; $p = 0.793$ for mean arterial BP; brachial site: $p = 0.255$ for systolic BP; $p = 0.611$ for diastolic BP; and $p = 0.372$ for mean arterial BP).

## 3. Discussion

It is estimated that 1.13 billion people worldwide suffer from hypertension, although less than half of the adults with hypertension are diagnosed and receive treatment [16]. In addition, elevated BP may cause vascular damage and accelerated conduit artery stiffening [17]. Here, we report in young adults that one night of forced wakefulness in a mainly seated position (i.e., mimicking sedentary occupations involving night shifts) increased arterial BP at rest the following morning. Adverse cardiovascular events, such as myocardial infarction, peak in the morning [5]. Thus, people with occupations involving night shifts might benefit from regular BP monitoring to anticipate possible adverse cardiovascular events.

Our finding that participants' BP was elevated after one night of forced wakefulness may partly explain why a meta-analysis of 27 observational studies demonstrated that

regular shift work is associated with 31% higher odds of hypertension [18]. For example, in a prospective cohort study of 2151 workers at manufacturing facilities, those with mostly night work and frequent rotations had a four-fold higher hypertension risk than non-night workers [19]. A previous study in six men suggested that elevated BP due to staying awake for one night may stem from baroreflex resetting toward higher BP levels [20]. In addition, the well-known suppression of the pineal gland hormone melatonin due to night work [21] may represent another potential mechanism underlying the association between night shift work and hypertension [1–3]. As suggested by both animal and human studies, melatonin, which is mainly released in the absence of environmental light, demonstrates significant hypotensive effects that can extend into the following day [22–25].

Our results showed that one night of forced wakefulness increased BP in men only. Thus, men appear to be more sensitive than women to the hypertensive impact of staying awake at night, at least in the short term. One explanation for why one night of forced wakefulness did not alter BP in women could be that they used hormonal contraceptives, including estrogen. This steroid hormone elicits vasodilatory effects and reduces sympathetic tone [26]. Furthermore, BP is inversely related to circulating estrogen concentrations during the menstrual cycle and is lower when 17β-estradiol levels peak [27]. Thus, more studies are needed to confirm our findings in freely cycling women.

Obesity has been established as a risk factor for elevated BP and hypertension [8]; however, whether people suffering from obesity may be more susceptible to the BP-increasing effects of staying awake for one night has not been addressed. Here, we show that young adults with normal weight appeared to be more susceptible to the impact of one night of forced wakefulness on BP than young adults with obesity. The difference was primarily seen for diastolic BP. One explanation could be that the hypertensive effects of one night of forced wakefulness may be less pronounced among people with obesity as they exhibit overall higher BP values, indicating a possible ceiling effect.

Several limitations must be considered in the interpretation of our findings. First, the observed effects of forced wakefulness on BP were only assessed in the morning. Given the circadian fluctuations of BP [4,28], our results may not necessarily extend to other times of the day. Additionally, it is unknown if daytime sleep following night shifts would have helped to normalize BP. It also remains unclear whether night shifts associated with high physical demands would produce similar effects on blood pressure, as seen herein after one night of forced sedentary wakefulness. Finally, to mimic night shift work, participants spent the sleep-deprivation night under normal indoor-light conditions (~500 lux). However, future studies applying constant routine protocols under dim light conditions are needed to decipher whether the observed rise in morning BP after one night of wakefulness is mainly due to sleep loss, nocturnal light exposure, or both.

## 4. Materials and Methods

### 4.1. Participants

Forty-seven young adults with either obesity (defined as a waist circumference >102 cm for men and >88 cm for women and a BMI $\geq$ 30 kg/m$^2$), respectively; N = 22) or normal weight (defined as a waist circumference <94 cm for men and <80 cm for women and a BMI < 25 kg/m$^2$, respectively; N = 25) participated in the study. They were non-smokers and free of medication, except for female participants (N = 21) who were using monophasic hormonal contraceptives before and during the study period. An in-laboratory screening session excluded acute and chronic illness, including a disease history interview and assessment of body measurements, hemoglobin A1c, and general blood status. All study participants had a habitual sleep duration of seven to nine hours between 22:00 and 08:00 during weekdays, had no extreme chronotype (MEQ score between 31 and 69 points; [12]), did not habitually nap during the daytime, had no employment involving night or rotating shifts, and did not travel across time zones in the three weeks before nor during the study period. The study was conducted according to the Declaration of Helsinki. The experimental procedures and expected outcomes of the study were reviewed and approved

by the Ethical Committee of Uppsala (DNR2017/560). All participants gave informed consent before the study onset. The results described herein are part of a more extensive study investigating the possible health consequences of one night of forced overnight wakefulness (e.g., [29,30]).

*4.2. Design and Procedure*

Between March 2018 and November 2020, participants took part in two conditions (one night asleep vs. one night awake), separated by about seven days. The order of experimental conditions was balanced across subjects. Within the week before the first experimental session, all participants spent one night asleep in our laboratory to avoid first-night effects on sleep in the experimental sleep conditions [31]. In the week before each experimental condition, subjects filled out sleep diaries to ensure adherence to a regular sleep schedule.

At 19:00, subjects received a standardized dinner upon arrival at the sleep laboratory. Afterward, no meals or caffeinated beverages were provided or allowed. In the sleep condition, ceiling room lights ($\approx$500 lux) were off between 23:00 and 07:00 the following day, providing participants a sleep opportunity of 8 h. Total sleep duration was determined by polysomnography via SOMNO HD (10–20 system; SOMNOmedics GmbH, Randersacker, Germany) according to American Academy of Sleep Medicine standard criteria ([32]; mean $\pm$ SD: 7 h, 12 $\pm$ 31 min). We also monitored desaturation events during sleep using the SOMNO HD pulse oximeter, defined as a decrease in the mean oxygen saturation of $\geq$3% for more than 10 s. Desaturation episodes were used to calculate the ODI, i.e., the mean number of desaturation episodes per hour [15]. In the forced wakefulness condition, subjects spent their time mainly sedentary with a selection of movies, games, and books and were continuously monitored by the experimenters.

In the morning, after sleep or forced wakefulness, BP was measured using an electronic sphygmomanometer (at ~08:30; Medisana BU510; Medisana GmbH, Germany) after 15 min of rest in a supine position at both the left and right ankles and upper arms, respectively. BP measurements were performed by experimenters following training sessions to standardize the procedures. Left and right ankle and brachial systolic and diastolic BP were averaged for the analysis. The mean arterial BP was computed using the following formula: average diastolic BP + 1/3 (average systolic BP- average diastolic BP).

*4.3. Statistical Analysis*

Data were analyzed using IBM SPSS Statistics 26 (SPSS Inc., Chicago, IL, USA). We applied generalized linear mixed models (GLMMs), assuming an unstructured covariance matrix and using a normal distribution (visually verified with histogram plots) with an identity link function to determine the effects of the following fixed factors on arterial BP outcomes: within-subjects factor experimental condition (one night asleep vs. one night of forced wakefulness), as well as participants' biological sex (men vs. women) and weight group status (participants with normal weight vs. those with obesity) as between-subjects factors. Subject-ID was included as a random factor. We also investigated possible interaction effects between the experimental conditions and participants' biological sex and weight group status (both interaction terms were included simultaneously in the GLMMs). Unless otherwise specified, data are reported as estimated marginal means $\pm$ standard error (SE). $p < 0.05$ was considered significant.

Two of the 47 study participants (both men with obesity) slept less than half of the eight hours in their experimental sleep condition. Thus, we did not include their BP measurements performed after sleep in the primary GLMM analysis. Nevertheless, the results remained the same when including these two participants (data not shown).

**5. Conclusions**

Our findings suggest that a single night of wakefulness, which occurs in occupations involving night shifts, yields higher BP values the following morning. Thus, night shift

workers might benefit from regular BP monitoring. In addition, the observed sex- and weight-specific effects of overnight wakefulness on BP warrant further investigation in larger cohorts, involving more extensive BP monitoring.

**Author Contributions:** Conceptualization, L.T.v.E. and C.B.; Data curation, L.T.v.E., E.M.S.M. and C.B.; Formal analysis, L.T.v.E., P.X. and C.B.; Funding acquisition, C.B.; Investigation, L.T.v.E., P.X., E.M.S.M., M.I. and J.E.; Methodology, L.T.v.E., E.M.S.M., M.I., J.E. and C.B.; Project administration, L.T.v.E.; Resources, C.B.; Supervision, C.B.; Validation, L.T.v.E., E.M.S.M., M.I. and J.E.; Visualization, C.B.; Writing—original draft, L.T.v.E. and C.B.; Writing—review and editing, P.X., E.M.S.M., M.I. and J.E. All authors have read and agreed to the published version of the manuscript.

**Funding:** This research was funded by the Novo Nordisk Foundation [NNF19OC0056777], Swedish Brain Research Foundation [FO2022-0254], and Folksam Foundation.

**Institutional Review Board Statement:** The study was conducted in accordance with the Declaration of Helsinki, and experimental procedures and expected outcomes of the study were reviewed and approved by the Ethical Committee of Uppsala (DNR2017/560; 24 January 2018).

**Informed Consent Statement:** Informed consent was obtained from all subjects involved in the study.

**Data Availability Statement:** Upon request from L.T.v.E. or C.B., the data can be shared with researchers.

**Conflicts of Interest:** The authors declare no conflict of interest related to the manuscript's content. The funders had no role in the design of the study; in the collection, analyses, or interpretation of data; in the writing of the manuscript; or in the decision to publish the results.

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
