# Peer review of "Effects of One Night of Forced Wakefulness on Morning Resting Blood Pressure in Humans: The Role of Biological Sex and Weight Status"

_2624-5175, doi:10.3390/clockssleep4030036_

Round 1
Reviewer 1 Report
Dear authors,
I carefully read your manuscript, which aims to evaluate blood pressure after a night of normal and deprived sleep, also considering sex and weight differences.
Even though the topic seems unexplored until now and somehow innovative, I found several weaknesses in the manuscript and the study protocol. The first impression is that you used data from a greater research study and publication (as it is written with citation 24), and you try to use all the available data and force to link them to shift workers. Indeed, you analysed a sleep-deprived night, which, in my opinion, does not fully represent a night shift worker. It could be supposed that a shift worker, who has been working for years at night, could have also developed some physiological adaptations not visible and different after a single night of sleep deprivation. The problem of shift workers is linked not only to working during the night, but also to the irregular times they must go to sleep and wake up they must respect during all shifts. For example, if they work the morning shift, they have to get up at five or earlier in the morning, which is very different from when they work the afternoon shift. This is just to say that your conclusions could be too reductive considering the complexity of medical night shift workers.
Especially in the discussion section, you sometimes confused or interchanged the sleep-deprived night with the night shift, for example, in lines 137-139.
Moreover, you evaluated blood pressure only after the wakeup; however, the blood pressure has a circadian rhythm, and who can assure that the higher blood pressure you registered is not a peak after the sleep offset and that it can decrease during the day? To be sure of your conclusions, you should have monitored blood pressure at several time points during the day and verified if it remained higher all day long.
Finally, women made use of contraceptives. A part of the fact that it is not clear if they already assumed it or if they started explicitly for the study (in this case, an adaptation period was necessary), but this made the two samples not superimposable, and the use of the contraceptive pill could invalidate your conclusions.
MINOR COMMENTS
1. TITLE: in the title, there are no mentions to shift work; however, the manuscript is based on night shift workers.
2. The introduction begins with the importance of the night shift; however, as previously said, you analysed a night of sleep deprivation. In my opinion, the introduction should be better contextualised to the analysis and the study protocol.
3. Results are difficult to read; why not insert more tables with the mean and comparison results?
4. Why did you write for SLEEP (for example, in line 64) after the data? What does it mean?
5. Line 71 seems more like a conclusion o a sentence for the discussion section, not for the results one.
6. I do not understand if the 2.3 section is a deepening of lines 72-75. Are these lines really needed? If yes, the results need to be better explained
7. 2.4: in my opinion, this section could be included in the statistical analysis section.
8. Chronotype assessment is not explained in the method section.
9. Did you take into consideration the weight or the obesity status, because sometimes you speak about obesity and sometimes about weight? Does weight reflect the obesity status? (probably not if you think about the lean and the fat-free mass)
Author Response
We would like to express our gratitude for the time the referee invested in improving our manuscript. All changes made to the manuscript are highlighted in yellow. Please find our point-by-point responses attached.

Reviewer 2 Report
Introduction - please expand this article with other important information/close to the topic of the paper. Also, highlight very clearly what was the novelty of this study, as well as its hypotheses.
Keywords: please delete: ''keyword 1; 2; 3;''.
''Forty-seven young adults with either obesity or normal weight''....ok, but what was the initial number of participants? was people who refused to participate in this study or did not fit into the established parameters? please complete with the necessary information for the best possible clarity of subject recruitment.
Even if it is not a main evaluation tool in this research ''Morningness Eveningness Questionnaire (MEQ)'' should be presented in more detail in the Methods chapter so that all the readers understand its role and content very clearly.
Author Response

(The authors gave the same response as above.)
